# Cutaneous Melanoma Systematic Diagnostic Workflows and Integrated Reflectance Confocal Microscopy Assessed with a Retrospective, Comparative Longitudinal (2009–2018) Study

**DOI:** 10.3390/cancers14030838

**Published:** 2022-02-07

**Authors:** Giovanni Pellacani, Francesca Farnetani, Johanna Chester, Shaniko Kaleci, Silvana Ciardo, Sara Bassoli, Alice Casari, Caterina Longo, Marco Manfredini, Anna Maria Cesinaro, Francesca Giusti, Antonio Iacuzio, Mario Migaldi

**Affiliations:** 1Dermatology Unit, Department of Surgical, Medical, Dental and Morphological Sciences related to Transplant, Oncology and Regenerative Medicine, University of Modena and Reggio Emilia, 41124 Modena, Italy; pellacani.giovanni@gmail.com (G.P.); johanna.chester@gmail.com (J.C.); shaniko.kaleci@gmail.com (S.K.); silvana.ciardo@gmail.com (S.C.); sarabassoli79@gmail.com (S.B.); alice.casari@gmail.com (A.C.); longo.caterina@gmail.com (C.L.); manfredini07@gmail.com (M.M.); francesca.giusti0505@gmail.com (F.G.); 2Department of Clinical Internal, Anesthesiological and Cardiovascular Sciences, Dermatology Clinic, Sapienza University of Rome, 00185 Rome, Italy; 3Centro Oncologico ad Alta Tecnologia Diagnostica, Azienda Unità Sanitaria Locale, IRCCS di Reggio Emilia, 42123 Reggio Emilia, Italy; 4Department of Anatomic Pathology, Azienda Ospedaliero—Universitario Modena, 41124 Modena, Italy; cesinaro.annamaria@policlinico.mo.it; 55th Engineer Regiment, Italian Army, Macomer, 08100 Nuoro, Italy; iacuzio.valore@gmail.com; 6Department of Anatomic Pathology, University of Modena and Reggio Emilia, 41124 Modena, Italy; mario.migaldi@unimore.it

**Keywords:** melanoma, number needed to excise, skin cancer, epidemiology, health services research, cancer patient pathways

## Abstract

**Simple Summary:**

A retrospective study (2009–2018), where the integration of education, a diagnostic-therapeutical workflow (with centralized and immediate assessment of suspicious lesions) and high-performance technology in a single province, seems to improve melanoma detection accuracy, as measured by the number needed to excise. From 40,832 lesions excised there was 279% increase number of melanoma excised. A systemic diagnostic-therapeutical workflow including reflectance confocal micrscopy (RCM) was described. The impact was assessed with the number of lesions needed to excise (NNE) according to excision location: reference hospital (DP) or other (NDP), NNE decreased almost 3-fold at DP and by half at NDP. Aggressive melanoma detection remained unchanged over the study period. Hypothetical cost analyses estimated health service savings.

**Abstract:**

Background: The increasing global burden of melanoma demands efficient health services. Accurate early melanoma diagnosis improves prognosis. Objectives: To assess melanoma prevention strategies and a systematic diagnostic-therapeutical workflow (improved patient access and high-performance technology integration) and estimate cost savings. Methods: Retrospective analysis of epidemiological data of an entire province over a 10-year period of all excised lesions suspicious for melanoma (melanoma or benign), registered according to excision location: reference hospital (DP) or other (NDP). A systematic diagnostic-therapeutical workflow, including direct patient access, primary care physician education and high-performance technology (reflectance confocal microscopy (RCM)) integration, was implemented. Impact was assessed with the number of lesions needed to excise (NNE). Results: From 40,832 suspicious lesions excised, 7.5% (*n* = 3054) were melanoma. There was a 279% increase in the number of melanomas excised (*n* = 203 (2009) to *n* = 567 (2018)). Identification precision improved more than 100% (5.1% in 2009 to 12.0% in 2018). After RCM implementation, NNE decreased almost 3-fold at DP and by half at NDP. Overall NNE for DP was significantly lower (NNE = 8) than for NDP (NNE = 20), *p* < 0.001. Cost savings amounted to EUR 1,476,392.00. Conclusions: Melanoma prevention strategies combined with systematic diagnostic-therapeutical workflow reduced the ratio of nevi excised to identify each melanoma. Total costs may be reduced by as much as 37%.

## 1. Introduction

The increasing global burden of melanoma (incidence, prevalence, mortality and morbidity) has increased the demand for efficient health care services to examine suspicious lesions [1,2,3,4,5]. Accurate early melanoma diagnosis enables curative, surgical excision, whilst delayed diagnosis has poor prognosis [6].

Projections suggest that due to increasing ultraviolet exposure and life expectancy, together with improved registration procedures and diagnostic tools [7], worldwide future melanoma incidence rates will keep on rising [8]. Australia is an exception, having slowed overall incidence rates, possibly due to prevention strategies [1,2,3,4,5,6,7,8,9]. Despite public education and preventive strategies increasing earlier diagnoses, it is debated if there is a corresponding favorable cost–benefit ratio when considering reduced melanoma morbidity and mortality [10,11].

Early diagnosis through prevention programs requires the coordination of public awareness, specialized screening of patients at high risk and appropriate diagnostic tools. From 2000, various local and national awareness programs for melanoma prevention have been undertaken, including population-based mass media campaigns for public education, such as the EUROMELANOMA [12] and “Skin Cancer Day”, which offered open access to skin cancer clinics [13].

Part of the rise in melanoma incidence may be due to overdiagnosis from increased awareness and screening [14,15]. The number needed to excise (NNE) evaluates diagnostic accuracy by calculating the number of biopsies needed to detect one skin cancer; a lower NNE implies higher precision. Estimates of NNE for benign lesions to find melanoma range between 20 and 30, with the exception of highly specialized centers (NNE ranges between 8 and 10) [16]. Thus, improved melanoma diagnostic ability in the context of screening is essential to reduce unnecessary biopsies, which increase the false positive rate, leading to unwarranted patient psychological stress and, secondly, an increased economic impact of biopsy evaluation [17]. 

Dermoscopy greatly improves diagnostic accuracy compared to naked-eye examinations [18,19]. However, a consistent burden to health services remains, and estimates depend upon medical expertise and available diagnostic tools [19]. Some specialized centers have adopted reflectance confocal microscopy (RCM), non-invasive cytological imaging of the skin, which enables the containment of this surgical burden through an excellent sensitivity and specificity compared to histology [19,20,21,22]. 

Accurate management through effective therapeutic–diagnostic pathways is essential to achieve a balance in patient care (waiting lists and delayed diagnoses) and health care system burden. In order to create clinical guidelines, robust strategies directed towards the measurement and monitoring of strategy implementation are essential [23]. Cost savings associated with the introduction of high-performance technology has been previously estimated [24].

This longitudinal study aims to assess the impact of the implementation of melanoma prevention strategies and a systematic diagnostic-therapeutical workflow of improved patient access and high-performance technology integration, by means of a retrospective analysis of epidemiological data collected for a single Italian province over a 10-year period in terms of NNE and estimated cost savings.

## 2. Methods

Data from the Department of Pathology, University of Modena and Reggio Emilia, Italy, including all cutaneous lesions with clinical suspicion of melanoma, excised within the province over a 10-year period (2009–2018), were obtained, extrapolated for clinical performance evaluation and clinical practice improvement. Local protocol specifies the complete excision of suspicious melanocytic lesions. Excision of non-melanoma skin cancers and lesion shavings (for example for cosmetic reasons) were excluded, thus only diagnoses of “melanoma”, “melanocytic nevus”, “seborrheic keratosis”, “solar lentigo” or “lichen-planus-like keratosis” were included. We classified lesions according to where the individual specimens were excised, either from the Department of Dermatology (DP) or any other health service provider within the province (NDP, including dermatologists in regional hospitals and visiting clinics and non-dermatology specialty clinicians). In Italy, lesions are excised by specialists and not primary care physicians. This study was approved by the local Ethics Committee (Prot. 3822/C.E.).

Province population data for the period of the study were obtained for reference [25]. Lesions were classified according to histopathological diagnoses, as either benign (BN, including melanocytic nevi, seborrheic keratosis, solar lentigo and lichen-planus-like keratosis) or melanoma (including in situ, superficial spreading and nodular melanoma, and lentigo maligna). All diagnoses were performed over the study period by the same dermo-histopathologist (A.M.C.) according to the same histopathological criteria; immunohistochemistry was introduced during the study period. Historical data for the same region previously estimated an incidence of melanoma over the period 1997–2004 of 9.7 for invasive melanoma only and 11.9, including in situ melanoma [26]. In the province, various prevention programs and a systematic diagnostic-therapeutical workflow were implemented, including: (i) “Skin Cancer Day” from 1999, to heighten patient awareness and encourage contact with dermatology specialists; (ii) the Provincial Regulatory document, approved in July 2013, enabling direct access for all patients with suspicious lesions referred by primary care physicians to same-day dermatological specialist services at DP; (iii) annual obligatory continuous education programs for primary care physicians from 2013 to 2016; and (iv) the integration of high-performance technology of RCM in DP in the diagnostic workflow in 2013 to enhance both sensitivity and specificity of atypical lesions assessment and the patient pathway for more immediate in vivo diagnostics [20]. The key implementations of the systematic diagnostic-therapeutical workflow employed throughout the study period and the associated outcomes are summarized in Figure 1. 

## 3. Statistical Analysis 

We grouped patients into 3 age groups (0–49, 50–70 and >70). The age-specific prevalence of melanoma was calculated with stratified age groups and overall standardized prevalence according to age and residence distribution of the 2009 provincial population data, using the direct method. 

The impact of the prevention and diagnostic strategies was assessed according to the total number of lesions excised/year, total number of melanoma excised/year and overall and annual NNEs. NNE was calculated as the ratio of total number of excisions to melanoma ([BN + melanoma] / melanoma). Data were analyzed according to source of lesion excision (DP or NDP) over time using the t-test. Melanoma Breslow thickness reported by the histopathologist was used to determine either a thin (<1 mm) or thick (≥1 mm) melanoma. 

The r^2^ coefficient of determination was used to assess identified trends: 0.81 < r^2^ < 1.00 high correlation corresponds with 0.51 < r^2^ < 0.8 moderate correlation and 0.5 < r^2^ < 0.0 low correlation. Analyses were conducted using STATA program version 14 (StataCorp LP 4905 Lakeway Drive College Station, Texas, USA). A value of *p* > 0.05 was considered significant.

Estimates for cost savings of the integration of high-performance technology.

Cost analyses were based on estimates previously published for the reference center, calculated over the period 2011–2015 [24]. As previously proven, within the province, there is an almost equal distribution of the diagnostic and surgical activities, all within a standardized and controlled medical process. In the current study, given that the total number of lesions evaluated at DP over the study period was unavailable, estimates were calculated based on the NNE at NDP, as follows: 

## 4. Results

During the 10-year study period a total of 40,832 lesions suspicious for melanoma in 33,732 patients were excised and analyzed in the province. Lesion diagnoses included 3054 melanomas (7.5% of excised lesions). During the same period, the provincial population increased by 2%, with aging of the population evident (inhabitants >50 increased by 13.5%). The standardized mean prevalence of melanoma increased by 4.4 per 10,000 inhabitants (Appendix A).

Benign lesions, according to frequency of observations, included nevi (*n* = 29,968; 73.4%), seborrheic keratosis (*n* = 5867; 15.5%), solar lentigo (*n* = 1195; 2.9%) and lichen-planus-like keratosis (*n* = 838; 2.2%). Excisions per year ranged from 3411 to 4749, with a clear trend towards increasing numbers of melanoma from 2014; the average number of melanoma/year prior to and following 2014 doubled (200 vs. 410). 

Despite the higher number of lesions excised, the ratio of melanoma identification improved; in 2009, 5.1% of all lesions excised were melanoma, whereas in 2018, this percentage had grown to 12.0% (see Table 1). 

The total NNE for the study period was 13.4 (average 15.1 (range = 8.3–22.6)) (see Table 1). Overall NNE for the province reduced (Figure 2, blue line), despite a 279% increase in the number of melanomas excised over the 10-year period (*n* = 203 (2009) to *n* = 567 (2018)) (see Table 1). 

Overall, most lesions were excised at NDP, but most melanomas were diagnosed at DP (see Table 2). 

There was an increase in the percentages of melanoma diagnoses among all lesions excised at both DP and NDP; correlation was high at DP (89%) and moderate at NDP (74%) (see Figure 3). 

At study initiation, DP excised fewer nevi for every confirmed melanoma (NNE for DP was 14 vs. NDP of 31; see Appendix A). From then, the NNE initially increased for both DP and NDP, presumably due to the effect of public and primary care physician awareness campaigns. Following the introduction of the new diagnostic-therapeutical workflow at DP, NNE decreased almost 3-fold compared to study initiation and remained low, despite the increasing number of melanomas diagnosed after 2014 (Figure 2, orange line/Appendix A). In NDP, the precision of lesion excision improved by more than half (Figure 2, green line), presumably associated with the transfer of most complex cases to DP. At study closure, the efficacy of melanoma lesion identification in terms of NNE for NDP had decreased to 12 (see Appendix A). The overall NNE was significantly lower for DP (NNE = 8) compared to NDP (NNE = 20), *p* < 0.001 (see Appendix A).

Throughout the study, the overall number of thin melanomas diagnosed in the province increased by more than 3 times, whilst the number of thick melanomas remained stable (see Table 1). Estimates highlight that the overall ratio of thin to thick lesions at study initiation was 3:1, which increased to 9:1 at the end of the study period. 

The percentage of thick melanomas in overall melanoma diagnoses throughout the study period decreased from 2014 (see Table 1). At DP the correlation was moderate (54%), whilst at NDP, it was low (18%) (see Figure 4). 

Hypothetical estimates for the total number of lesions assessed at DP was 32,560 lesions (1628 MM × 20 NNE at NDP). Based on average costs previously calculated in a scenario with routine integration of RCM (EUR 104.50) [24], total cost estimates for the 10-year period at DP amounted to EUR 3,402,520 (32,560 assessed/excised lesions x EUR 104.50). Based on average costs previously calculated in a scenario of routine melanoma screening (EUR 143.63) [24], total cost estimates for the 10-year period at NDP were EUR 4,667,801 (27,855 excised lesions x EUR 143.63). The total cost saving of EUR 1,265,281 represents a 37% total cost reduction.

## 5. Discussion

This retrospective, longitudinal study, including all lesions excised in a province over a 10-year period, in the context of combined melanoma awareness, improved patient access and high-performance technology implementation into the clinical diagnostic pathway and highlights improved diagnostic accuracy within a population of minimal change. Further, both increased early diagnoses and potential benefits of centralized diagnoses of suspicious lesions were evident. These changes are expected to minimize the disease burden of health care systems. 

It has been estimated that melanoma prevention and early detection can offer strong health and economic benefits [27]. However, there seems to be a disparity regarding the efficacy and quality of dermato-oncological prevention strategies and care, with relative impacts scarcely measured [23]. Melanoma registries are often limited by the registration of selected cases, often biasing the quality and coverage of data availability, contributing to heterogeneity reported around the world [28]. This study includes a collection of complete data: all excised cutaneous lesions for histopathological assessment at a single pathology department within a geographical zone. It offers a unique opportunity to evaluate an integrated approach of improved public and first-level medical consultancy awareness and a systematic diagnostic-therapeutical workflow, not only within the center of reference, but also in the community as a whole. 

Educating awareness and mobilizing high-risk patients to timely dermatological assessment is essential to curtailing melanoma mortality. Increased patient and first-level medical practitioner awareness are probably responsible for the increase in the identification of suspicious lesions excised throughout the province. 

Directing skin biopsies to centers with favorable NNE may substantially decrease the economic burden of skin cancer on health care systems [29] and improve clinical outcome [17,30]. Data obtained in this longitudinal study support the call for centralized assessment of suspicious lesions.

However, Marushchak et al. suggested that the high cost of screening and potentially increased rates of lesion excisions, as well as overimaging and overtreating, pose serious concerns [31]. The integration of RCM in the diagnostic patient pathway improves diagnostic power for melanoma detection through the addition of morphologic clues, thereby reducing unnecessary excisions compared to dermoscopy [20]. The increased precision in melanoma detection in this study compares very favorably with estimates from dermatologists around the world, ranging from 12 to 16% [29]. Further, estimates from this study suggest that the integration of high-performance technology may improve melanoma detection by up to 37%.

The greatest increase in observed melanoma incidence is registered with in situ diagnoses, followed by thin, invasive melanoma [14,15]. By contrast, thick melanoma incidence has remained largely constant [14]. In the current study, the proportion of thin melanoma diagnosed over the study period increased by over 3 times, which is likely due to a combination of factors, including a true rise in sun exposure and increased detection rates [32]. 

For cancer screening to be effective, it must detect more cancers at an early stage and must lead to a lower incidence of late-stage disease over time [33]. However, in the current study, the number of thick melanoma lesions remained largely constant over the 10-year observation period. This study cannot resolve the doubt of whether the increase in early-stage disease detection represents effective prevention in controlling the number of thick (and thus aggressive) melanomas despite a real incidence increase, or whether it represents excessive diagnosis of lesions that may never have become invasive or caused death [15]. While melanoma tumor thickness is correlated with prognosis, we cannot be certain that earlier detection through screening will change prognosis [29]. Further data monitoring within a unique geographical area, where all excised lesions are analyzed at a single histopathological laboratory, will continue to shed light on whether the increased diagnosis of thin melanoma impacts the observation of more advanced stages of the disease. 

Overdiagnosis is difficult to measure, as practically all malignant lesions are excised, potentially leading to overtreatment (healthy people are exposed to unnecessary surgery and possibly adjuvant therapy). Overtreatment unnecessary labels healthy people with a suspect of a cancer diagnosis [15] and may appear to improve efficiency whilst actually leading to greater health costs. The integration of RCM proved to decrease the number of patients awaiting unnecessary diagnostic referrals for benign lesions by 30%, thereby reducing anxiety levels of patients awaiting histopathological diagnoses following surgical excision. The authors hypothesize that the prevalence of nevi diagnosed without excision probably constantly increased, but confirmatory data are unavailable. The only data available to assess diagnostic accuracy is therefore NNE. 

Our longitudinal study has several further limitations. Data suggests that dermatologists’ clinical diagnostic ability improves with experience [30], and prior to changes in workflow and where improved technologies were not introduced (NDP; dermoscopy use only), melanoma detection improved. NNE does not give any insight into the number of melanomas that go unbiopsied and, therefore, undiagnosed. Further, laboratory and provincial mortality data did not include the date of the first diagnosis so incidence could not be estimated. Economic estimates should be interpreted with caution. Budgetary information for the costs of health promotion or screening, and costs associated with the integration of high-performance technology had to be estimated for the total number of lesions screened at DP but not excised, which would have been excised without the integration of RCM. The impact of the introduction of IHC in histopathological diagnostics could not be assessed in this study but has been estimated elsewhere to change diagnoses in a very small portion of lesions [34]. Finally, the results obtained in this study are not generalizable to all practice settings.

## 6. Conclusions

The combination of melanoma prevention strategies and a systematic diagnostic-therapeutical workflow, with streamlined patient access and high-performance technology integration, proved an increase in more accurate in vivo diagnoses of thin but not thick melanoma, reducing the ratio of nevi excised for each melanoma identified for the whole province. Further longitudinal data are required to address doubts associated with overdiagnosis and overtreatment, when the effect of early diagnosis on thick lesions may be evaluated.

## Figures and Tables

**Figure 1 cancers-14-00838-f001:**
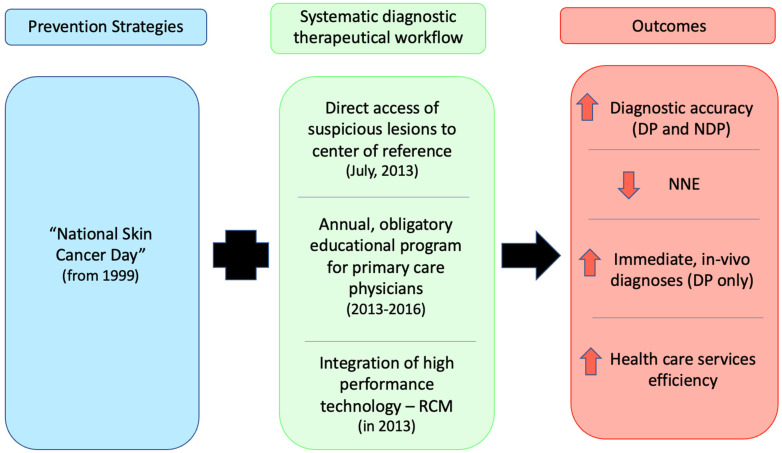
Flow chart of the introduction of intervention strategies over the study period.

**Figure 2 cancers-14-00838-f002:**
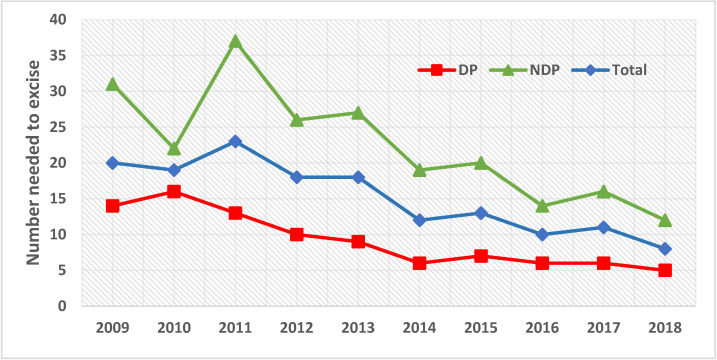
Trends of the number needed to excise for DP (red line), NDP (green line) and the total province (blue line) over the 10-year study period.

**Figure 3 cancers-14-00838-f003:**
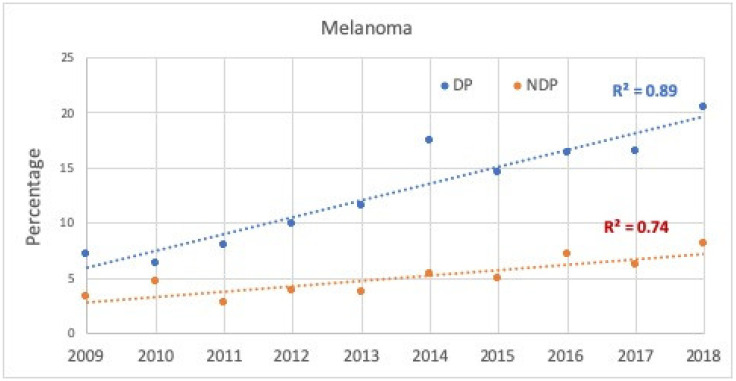
Proportion (%) of melanoma diagnoses for all suspected lesions excised during the study period at the Department of Dermatology (DP) or all other health institutions in the province (NDP), according to year of diagnosis.

**Figure 4 cancers-14-00838-f004:**
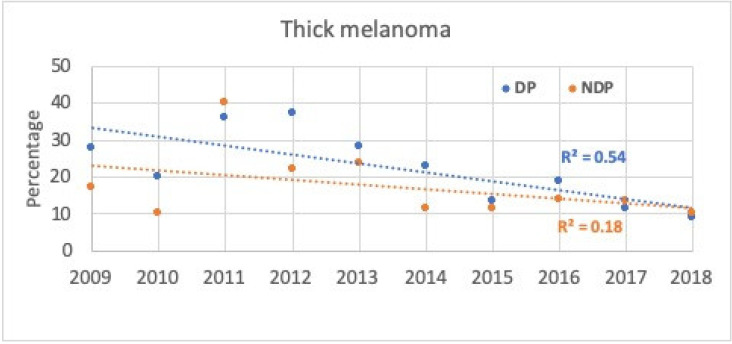
Annual proportion (%; dot) and linear trend (line) of thick melanoma diagnoses (>1 mm) for all melanoma lesions excised during the study period at the Department of Dermatology (DP) or all other health institutions in the province (NDP), according to year of diagnosis.

**Table 1 cancers-14-00838-t001:** All excised lesions during the study period according to year of excision. Lesions are also presented according to diagnosis (melanoma or melanocytic nevi) and overall number needed to excise and all melanoma lesions according to Breslow thickness (think <1 mm or thick >1).

	All lesions							Melanoma Thickness			
Year	Total		Melanoma		Nevi		NNE	Thin		Thick	
	*n*	% (total)	*n*	% *	*n*	% *		*n*	% ^	*n*	% ^
2009	4013	9.8	203	5.1	3810	94.9	20	154	75.9	49	24.1
2010	4300	10.5	225	5.2	4075	94.8	19	192	85.3	33	14.7
2011	4133	10.1	183	4.4	3950	95.6	23	114	62.3	69	37.7
2012	3718	9.1	201	5.4	3517	94.6	18	143	71.1	58	28.9
2013	3422	8.4	189	5.5	3233	94.5	18	140	74.1	49	25.9
2014	3653	8.9	309	8.5	3344	91.5	12	255	82.5	54	17.5
2015	3865	9.5	306	7.9	3559	92.1	13	267	87.3	39	12.7
2016	4267	10.5	429	10.1	3838	89.9	10	359	83.7	70	16.3
2017	4749	11.6	442	9.3	4307	90.7	11	387	87.6	55	12.4
2018	4712	11.5	567	12.0	4145	88.0	8	513	90.5	54	9.5
Total	40,832	100	3054	7.5	37,778	92.5	13	2524	82.6	530	17.4

NNE, number needed to excise. * Percentage calculated according to total lesions excised each year. ^ Percentage calculated according to total number of melanoma excised each year.

**Table 2 cancers-14-00838-t002:** Overall lesions excised over the study period. Number needed to excise and melanoma according to lesion thickness for the Department of Dermatology (DP) and other health institutions in the province (NDP).

	DP		NDP	
	No. Lesions	%	No. Lesions	%
No. all lesions:	12,977	31.8	27,855	68.2
Melanoma	1628	12.5	1426	5.1
Thin	1315	80.8	1209	84.8
Thick	313	19.2	217	15.2
Nevi	11,349	87.5	26,429	94.9
	Ratio		Ratio	
NNE	8:1		20:1	
NNE, DP:NDP *	1:2.5			

NNE, number needed to excise. * *p* <0.001.

## Data Availability

Data will be made available following any reasonable request sent to Francesca Farnetani.

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
