# Peer review of "Cutaneous Melanoma Systematic Diagnostic Workflows and Integrated Reflectance Confocal Microscopy Assessed with a Retrospective, Comparative Longitudinal (2009–2018) Study"

_cancers, 2022, doi:10.3390/cancers14030838_

Round 1

Reviewer 1 Report

Pellacani et al analyzed all skin biopsies from a province of Italy over a 10-year period (limited to melanoma, nevus, seb K, solar lentigo, LPLK). They identified that the proportion of biopsied lesions that were melanoma increased over the period. They suggest that a “systematic diagnostic-therapeutical workflow” was responsible for this trend. This study is interesting but, in my opinion, methodologically flawed.

  • The authors report the absolute numbers of biopsied melanomas and nevi in the province. It would be helpful to report the age-standardized incidence rates for these measures. The denominator of the population is important to interpreting this data. Did the population change in the province over time? More people? Got older (or younger)? More/fewer immigrants?
  • It is not clear that diagnostic accuracy actually improved; if the true incidence rate of melanoma doubled, NNE would be expected to decrease (as PPV is intrinsically associated with prevalence of disease).
  • If diagnostic accuracy improved over time, true incidence remained stable, and the population structure did not change, one would expect a decline in nevus biopsies. This did not occur, so it is likely that multiple factors are at play. Can the data be reported in a more descriptive way with less speculation?
  • If the workflow was responsible for earlier diagnoses of melanoma, one would also expect that the incidence of thick melanoma would decrease. This does not appear to be the case, as the rise in incidence appears to be driven by in-situ and thinly invasive melanomas. A more likely cause of their observations is that overdiagnosis increased over time (either via more screening or changing thresholds for labeling early disease); this gives the appearance of improved “efficiency” but it may actually be leading to greater costs.
  • Although it appears that the pathologist remained the same over the 10-year period for all lesions biopsied in the province, it is unclear if the performance and practice of the pathologist has changed. For example, was any new diagnostic technology introduced (i.e., IHC, FISH, CGH, etc.). It would be interesting to see the intra-observer reliability of the pathologist for a random selection of cases from the earlier years.
  • The observation that NNE decreased by ~50% in both DPs and NDPs further argues that a change in incidence due to overdiagnosis is probably responsible (not improved diagnostic accuracy due to technology like RCM, which should only affect the DPs).
  • Also most of the decline in NNE in DP occurred from 2009-2014 (prior to implementation of RCM in 2013).

The authors try to associate many of their findings with specific effects. As this is an observational study, it is not clear what exactly led to their data (i.e., true rise in incidence, change in diagnostic thresholds, more screening, etc.). If data from another province (in which the workflow was not introduced) could be compared to data from this province, that would strengthen the authors interpretation.

Specific points:

Line 177 – per Figure 2, the NNE at baseline for NDP appears to be 20 (not 31).

Line 51 – the reasons for the slowed incidence rates in Australia are not precisely clear. Please reword (i.e., possibly due to prevention strategies”

Line 52 – earlier diagnosis may save lives or it may not, this has not been proven for melanoma. please reword.

Figure 1 – the outcomes include “diagnostic accuracy” and “immediate in-vivo diagnoses”. What data supports the increase in diagnostic accuracy? Why is “in vivo diagnoses” included? What diagnoses were made soley by RCM (without biopsy) and where is the data to support those measures over time?

Line 64 – a lower NNE implies greater PPV/precision but this may not always be due to improved diagnostic accuracy. For this to be true, there must be no change in incidence and no change in threshold for biopsy, etc.. A limitation of analyzing NNE trends over time is that it is not clear if these assumptions are true, and they are not likely true over this period for this province.

Line 134-140. I don’t find the methodology for estimates of cost saving to be compelling or valid, without understanding the true denominator of persons living in the province. Either I would remove this or change.

Author Response

In atatch the file 

Reviewer 2 Report

  • Line 20: number of what? – word missing
  • Line 23: leave space between sentences and full stop is missing
  • Line 30: define NDP
  • 5% melanoma diagnosis of 40832 excisions seems relatively low especially given the fact that in Italy, lesions are excised by specialists (line 97). I would expect the % of correct diagnosis to be much higher for specialists. Is this detection rate comparable to other countries? In Australia, the detection rate is around 10% for GPs, not specialists.
  • Line 34: what points do you compare to calculate the % increase in melanomas excised?
  • Line 201: does the cost estimate take into consideration the need for subsequent wider excisions. Does this include cost estimates for histopathological analysis of simply excision?
  • Line 205: the reported cost savings are encouraging. However, do you think these savings would be offset by the costs involved in the role out of RCM and the subsequent training requirements for this technology over the same time period?
  • The quality of the images is poor when printed. These may need to be saved and inserted at a higher resolution?

Author Response

in attach the file 

Reviewer 3 Report

This study assessed the impact of a systematic diagnostic-therapeutical workflow and found the number of skin excisions needed in order to diagnose melanoma has decreased with increased diagnostic accuracy and reduced costs. It is an interesting study.

How was the implementation of the systematic diagnostic therapeutical workflow evaluated in the Department of Dermatology (DP) and other (NDP) respectively? What is the percentage of dermatologists or physicians trained to use RCM in DP and NDP respectively?

Were all the specimens excisions? Are there any shave or punch biopsies, especially biopsies for benign lesions without follow up excision? Are duplicates of the same lesion removed from the study (ie shave biopsy followed by excision). Please clarify.

Labels of each column in table 2 are missing therefore it is difficult to interpret the numbers. 

Author Response

in attach the file 

Round 2

Reviewer 1 Report

Thank you for your revisions to my suggestions. Unfortunately, in this current form, I continue to find significant methodological limitations and question the validity of the study findings.